

# How tapeworms interact with cancers: a mini-review

Manfred Schreiber, Vojtěch Vajs and Petr Horák

Department of Parasitology, Faculty of Science, Charles University Prague, Prague,
Czech Republic

## ABSTRACT

Cancer is one of the leading causes of death, with an estimated 19.3 million new cases
and 10 million deaths worldwide in 2020 alone. Approximately 2.2 million cancer
cases are attributed to infectious diseases, according to the World Health
Organization (WHO). Despite the apparent involvement of some parasitic helminths
(especially trematodes) in cancer induction, there are also records of the potential
suppressive effects of helminth infections on cancer. Tapeworms such as
*Echinococcus granulosus*, *Taenia crassiceps*, and more seem to have the potential to
suppress malignant cell development, although in a few cases the evidence might be
contradictory. Our review aims to summarize known epidemiological data on the
cancer-helminth co-occurrence in the human population and the interactions of
tapeworms with cancers, *i.e.*, proven or hypothetical effects of tapeworms and their
products on cancer cells *in vivo* (*i.e.*, in experimental animals) or *in vitro*.
The prospect of bioactive tapeworm molecules helping reduce the growth and
metastasis of cancer is within the realm of future possibility, although extensive
research is yet required due to certain concerns.

## INTRODUCTION

Infections with pathogens have been proposed as one of the possible triggers of cancer
development (*Colotta et al., 2009*). However, there is also some evidence of cancer
suppression due to concurrent infections (*Oikonomopoulou et al., 2013*). Although the
main emphasis is on viruses and bacteria (*Schiller & Lowy, 2021*; *Duong et al., 2019*),
eukaryotic parasitic infections cannot be omitted. Parasites such as *Leishmania* spp. (*Caner
et al., 2020*; *Al-Kamel, 2017*), *Trypanosoma cruzi* (*Ribeiro Franco et al., 2023*), and
*Toxoplasma gondii* (*Baird et al., 2013*) are hypothesized to be able to influence cancer
development; their attenuated forms or products show a promising anti-cancer effect *in
vivo*/*in vitro*. Regarding helminths as multicellular parasites, the research of
cancer-pathogen interactions is still in its infancy.

Helminths are prevalent parasites in humans, mainly in tropical and subtropical
countries. For example, it is estimated, that the number of people infected by
soil-transmitted helminths reaches up to 1.5 billion (*Fong & Chan, 2022*). On the other
hand, in high-income global north countries with high hygiene standards, the number
of human infections by helminths is usually low. The latter situation might have

Corresponding author
Petr Horák,
petr.horak@natur.cuni.cz

unanticipated consequences: the absence of certain pathogens, including helminths, may result in a more frequent occurrence of other (mainly autoimmune) diseases, which are rare in helminth-rich communities ("hygienic hypothesis" and "old friend hypothesis" (*Strachan et al., 2000*; *Rook, 2010*)). In other words, helminths in humans influence the immune system by exposing it to many antigens (in a sort of training) and by producing bioactive molecules that modify specific immune reactions. Therefore, helminths are well pre-adapted to form a tight association with their human hosts.

Not only is the immune system influenced by helminths (in many cases, modified T-helper cell 2 (Th2) and regulatory T cell (Treg) responses are triggered in chronic helminthoses (*Maizels & Yazdanbakhsh, 2003*; *Hewitson, Grainger & Maizels, 2009*)), but these infections may impact other types of human diseases, namely cancers. Some helminths are known as proven cancerogenic agents (*Clonorchis sinensis*, *Opisthorchis viverrini*, *Schistosoma haematobium*; these flukes belong to group 1 human carcinogens as of 2012 (*Bouvard et al., 2009*)). Human cancers caused by helminth infection may include cholangiocarcinoma, colorectal cancer, hepatocellular carcinoma, urinary bladder cancer, and other malignancies (*e.g.*, *Scholte, Pascoal-Xavier & Nahum, 2018*; *Correia da Costa et al., 2021*; *Wu et al., 2022*). On the other hand, some helminths display possible cancer-suppressing ability which has been indicated by some epidemiological surveys for humans, animal experiments or *in vitro* (see below for details). For example, in the mouse model, *Trichinella spiralis*, as a representative of nematodes, has been repeatedly associated with cancer suppression (*Wang et al., 2009*; *Kang et al., 2013*; *Vasilev et al., 2015*). In the human population, some epidemiological data support this view. For example, the decline of *Ascaris lumbricoides* prevalence in Korea during 1971–1992 has been followed by a remarkable increase in breast cancer incidence during 1992–2013 (*Yousefi et al., 2023*) and the mean survival time of patients with adult T-cell leukemia was longer if they had a concurrent infection with *Strongyloides stercoralis* (*Plumelle et al., 1997*). The first possible cancer-suppressing effect of a flatworm has been postulated with *Echinococcus granulosus*, a highly pathogenic cestode responsible for human cystic echinococcosis (*van Knapen, 1980*).

Regarding tapeworms (Cestoda), their life cycles contain one or two intermediate hosts and a final host. Based on the tapeworm species, humans can harbor either larval or adult cestodes (or both, like in the case of *Taenia solium*). The adults live exclusively in the digestive tract (the intestine), whereas larval stages invade various tissues/organs and may be life-threatening (larvae of *Echinococcus* spp., *T. solium*, *etc.*). Although some experimental data with laboratory rodents (see below) show that the presence of tapeworm larvae or their products could negatively influence cancer cells, it is ethically and practically disputable to infect cancer-bearing patients with living worms, which cause other pathologies. In this regard, the situation substantially differs from some other helminth treatment procedures considered in the recent years (*e.g.*, the curative application of *Trichuris suis* in people with inflammatory bowel disease (*Huang et al., 2018*)). Fortunately, state-of-the-art parasitology offers advanced molecular tools to characterize and produce parasite effector molecules *in vitro*. Such molecules can potentially affect

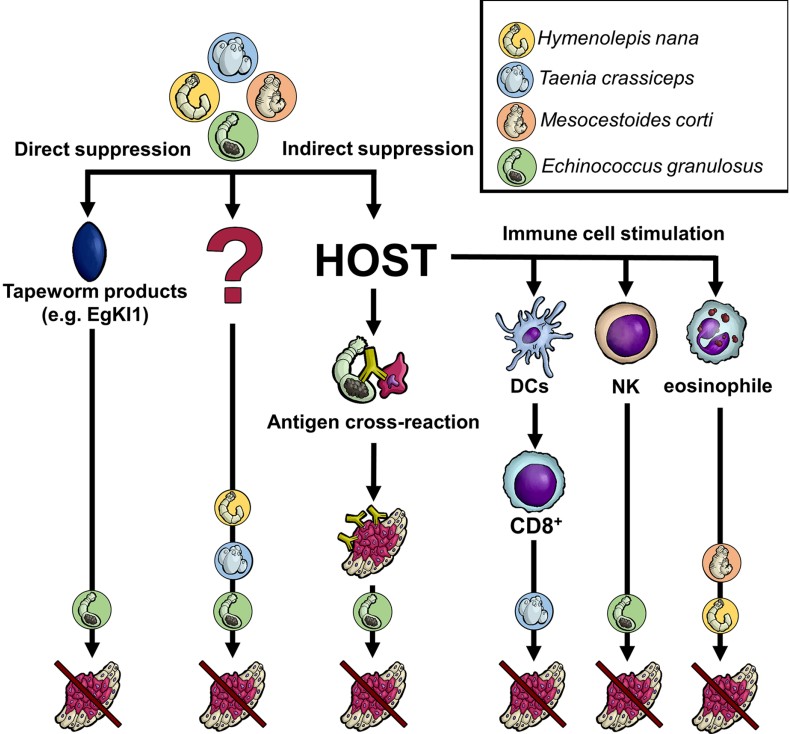

**Figure 1 Anticipated mechanisms of tapeworm effects on cancer.** Although the pathways by which tapeworms suppress tumor development are unknown in many cases, several proposed mechanisms exist. The worms could directly damage cancer cells through their products or indirectly by influencing the host immune system. The latter may occur either due to the parasite exhibiting the same antigen epitopes as cancer cells or because its products affect various immune system cells, directly killing cancer cells or activating other immune cells. Of course, some not yet recognized processes/interactions might also be involved. DCs–dendritic cells, NK–natural killer cells, CD8+–CD8+ T cells, EgKI1-Kunitz-type protease inhibitor produced by *E. granulosus*.

cancer cells directly (tapeworm molecules interfere with the cancer cell activities) or indirectly (stimulation of the immune system to intervene in cancer growth/survival) (Fig. 1).

Our review aims at the up-to-date knowledge of *in vivo* (using experimental animals)/*in vitro* interactions between tapeworms/tapeworm products and cancers, reflecting, for particular helminth species and infections caused by them, the circumstantial epidemiological pieces of evidence, *in situ* observations as well as experimental data using the tools of cell biology.

## SURVEY METHODOLOGY

Our study includes articles on tapeworms and their effect on cancers. Due to the complexity of helminth-influenced immune responses, we included only studies discussing the connection between tapeworm-activated immune systems and cancer. The Web of Science and PubMed databases were used for the literature search. The following keywords and their combinations were used: "Tapeworm", "Cestod", "Cancer", "*Taenia*", "*Echinococcus*", "*Mesocestoides*", and "*Hymenolepis*"; the alternative terms like "Echinococcosis" or "Hydatidosis" were also considered. The articles were checked for

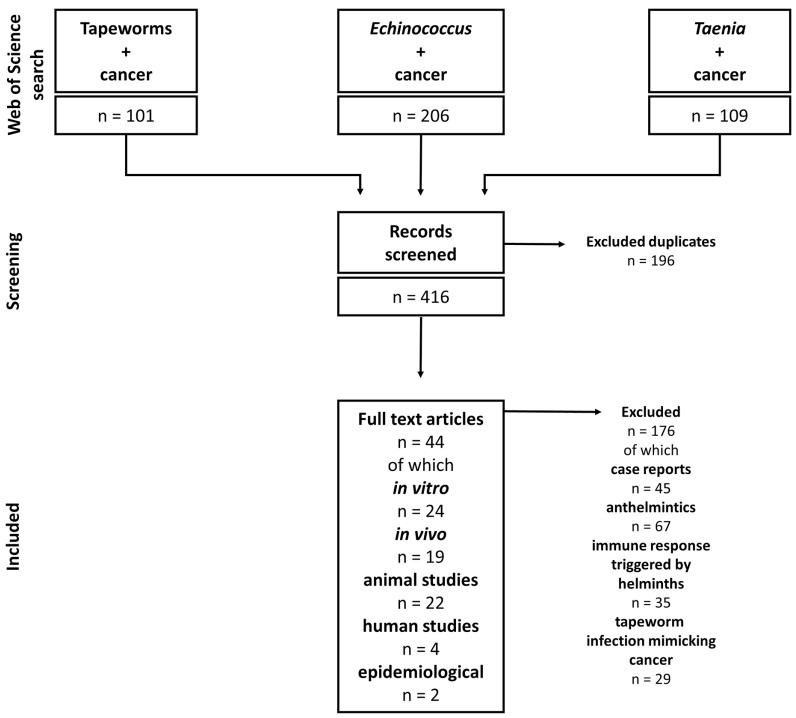

**Figure 2  Flowchart of how the literature was selected.**

their suitability for this review. The non-relevant articles (covering, *e.g.*, anthelmintic drugs and their effect on cancer, and immune responses triggered by helminth infections with no relation to cancer) were excluded. On the other hand, all articles describing any cestode-cancer and cestode-immune system-cancer interactions were selected and analyzed. For example, the combination "cancer" AND "tapeworm" (Web of Science) produced 101 results, and 39 were found reliable and used in our review. Altogether, we included 44 papers on cestode-cancer interactions in our review (Fig. 2).

## Particular tapeworm species and cancer

### *Echinococcus granulosus*

Several epidemiological studies indicated a possible influence of *E. granulosus* cysts on cancers in humans. *Akgül et al. (2003)* observed a reduced cancer prevalence in *E. granulosus* patients in a retrospective study in Turkey, and the authors hypothesized that the infection could suppress cancer development. Although there is also a retrospective study from Cyprus with results to the contrary (*Oikonomopoulou et al., 2016*). However, the latter involved patients with previously treated cystic echinococcosis and not an active infection at the time of cancer diagnosis (*Gundogdu, Saylam & Tez, 2017*). Another retrospective study showed that patients with hepatocarcinoma co-infected with *E. granulosus* had a longer survival time than those without this tapeworm (*Bo et al., 2020*).

Recently, the interactions of *E. granulosus* with host immunity and cancers have been thoroughly reviewed by *Guan et al. (2019)*. As to the experiments with living larvae, the

promising cancer-suppressing effects were tested on fibrosarcoma cells *in vitro*. Co-cultivation with tapeworm larvae led to the inhibition of proliferation and reduced viability of the cells (*Darani et al., 2012*). Additionally, *in vivo*, laboratory rats infected with *E. granulosus* protoscoleces showed significant suppression of chemically-induced mammary carcinoma development (*Altun et al., 2015*).

Due to the pathogenicity of this tapeworm, the research has frequently shifted toward selected components/fractions and identified molecules of *E. granulosus* larvae. Injection of larval cystic fluid into mice, which were then inoculated with B16F10 melanoma cells, resulted in tumor growth inhibition (*Darani et al., 2016*); the same happened if the cystic fluid was applied to mice with already developed melanomas (*Rad et al., 2018*). Regarding particular components, the 78 kDa fraction was confirmed to produce a similar effect in mice (*Rad et al., 2018*). As for other types of cancers, the cystic fluid triggered the inhibition of tumor growth in mice injected subcutaneously with CT27 colon carcinoma cells; the survival time of mice significantly improved (*Berriel et al., 2013*). Similar results were also observed using a lung cancer line (*Berriel et al., 2021*).

Under *in vitro* conditions, apoptosis of breast cancer cells was induced in the presence of *E. granulosus* larval cystic fluid. This effect was also observed using glycoprotein fractions and one 78 kDa fraction of the cystic fluid (*Daneshpour et al., 2019*). As for melanomas, the miRNA-365 component of the cystic fluid can induce apoptosis in the A375 melanoma cell line (*Mohammadi et al., 2021*); however, this conflicts with the results from *Gao, Zhang & Huang (2018)*, who described inhibition of apoptosis using the complete cystic fluid.

Regarding particular bioactive molecules, EgKI-1 is a Kunitz-type protease inhibitor produced by *E. granulosus* (*Ranasinghe et al., 2015*). Culturing several human tumor cell lines *in vitro* in the presence of recombinant EgKI-1 inhibited their growth and hindered the ability of these cells to migrate, while not affecting the growth of normal cells. EgKI-1 was also able to induce apoptosis in human breast cancer cells. Applying this protease inhibitor to mice with the same breast cancer cell line also suppressed tumor growth *in vivo* (*Ranasinghe et al., 2018*). Another interesting Kunitz-type protease inhibitor from *E. granulosus* is Kunitz4 (EgKI-4). Peptides derived from Kunitz4 have been shown to induce apoptosis and inhibit proliferation in cancer cell lines *in vitro* due to being an ion channel blocker (*Rashno et al., 2023*).

Besides these direct effects of tapeworm products, immune-mediated processes may play a significant role. One of the critical processes in activating anti-tumor immunity is the recognition of antigens on the surface of tumor cells. Similarities between the antigens in the cystic fluid of *E. granulosus* and those produced by lung cancer cells were described (*Yong, Heath & Savage, 1979*). One of these antigens is the O-glycosylated Tn antigen (α N-acetylgalactosamine-O-serine/-threonine), expressed by larvae and adults of *E. granulosus*, which has also been detected in the sera of patients with cystic echinococcosis. This glycoprotein is abundant on the surface of many tumor cells, including lung, breast, and pancreatic cancers (*Springer, 1997*). Antibodies raised against *E. granulosus* larvae reacted with tumor cell excretory-secretory products. Analogously, the same cross-reactivity was observed with sera from breast cancer patients and *E. granulosus*

larval cyst antigens. Thus, it is thought that this antibody response against the Tn antigen could activate anti-tumor immunity (*Alvarez Errico et al., 2001*). Sera from breast cancer patients also responded to a non-glycosylated 27 kDa molecule isolated from *E. granulosus* larval cysts (*Sharafi et al., 2016*). Moreover, murine antibodies raised against *E. granulosus* cystic fluid recognize some antigens on CT27 colon carcinoma cells. Thus, the cross-reactivity of antigens could inhibit CT27 tumor growth in mice treated with *E. granulosus* cystic fluid (*Berriel et al., 2013*). An antibody cross-reaction could also explain the cytotoxic effect of serum from echinococcosis patients on human lung cancer cells *in vitro* (*Karadayi et al., 2013*). Regarding the cross-reactivity phenomenon, the cancer cell-derived antigens are believed to be poorly immunogenic. On the other hand, the parasite antigens are usually highly immunogenic; therefore, in the case of similar motifs shared between cancer and parasite, the cross-reacting antibodies might represent a path for cancer immunotherapy in the future (*Ubillos et al., 2007*; *Yousefi et al., 2023*).

*Echinococcus granulosus* may also affect specific groups of immune cells associated with anti-cancer immunity. For example, the application of a mucin-like peptide isolated from *E. granulosus* larvae (Egmuc) led to an increase in the number of activated NK cells in the mouse spleen, with NK cells being an essential component of anti-tumor immunity and used in tumor immunotherapy (*Salagianni et al., 2012*); activated NK1.1 cells probably also caused reduced tumor development in mice treated with *E. granulosus* cystic fluid (*Berriel et al., 2021*). Splenocytes isolated from Egmuc-treated mice had a cytotoxic effect on pancreatic tumor cells of the Panc02 line when cultured *in vitro* (*Noya et al., 2013*). In addition, EgKI-1 produced by *E. granulosus* inhibits neutrophil chemotaxis (*Ranasinghe et al., 2015*), and antigen B isolated from the cystic fluid of *E. granulosus* has the same effect (*Shepherd, Aitken & McManus, 1991*; *Mamuti et al., 2006*). Neutrophils are involved in the host immune response against *E. granulosus* infection (*Zhang, Ross & McManus, 2008*). They are also associated with the progression, metastasis, and angiogenesis in the tumor microenvironment (*Coffelt, Wellenstein & De Visser, 2016*). Thus, inhibition of neutrophil chemotaxis mediated by *E. granulosus* products could contribute to the observed anti-tumor effect of this tapeworm.

Although most of the research points to the cancer-suppressing effect of *E. granulosus*, there are also studies opposing this fact. For example, *Turhan et al. (2015)* postulate that *E. granulosus* infection in mice enhances the development of liver metastases with 4T1 cell line breast tumors. Furthermore, the co-cultivation of HepG2 cells with *E. granulosus* protoscoleces led to increased proliferation *in vitro* and larger subcutaneous tumors in mice (*Yasen et al., 2021*). It appears, then, that the products do not affect all cancer cell lines equally. Therefore, the story of *E. granulosus*-derived components participating in the fight against cancer must be taken with caution, and further studies are welcome.

### *Taenia crassiceps*

The tapeworm *T. crassiceps* is well known for its ability to rapidly reproduce in the intermediate host, making it a suitable laboratory model using rodents (*Willms & Zurabian, 2010*). As for humans, infections by larval stages are quite rare and appear to be linked with immunocompromised patients (*Heldwein et al., 2006*); therefore, there is no

epidemiological data on tapeworm-cancer associations in humans. The direct effect of *T. crassiceps* cysticerci on cancer was investigated in the experimental model of ulcerative colitis-associated carcinoma (CAC). Infection of mice with *T. crassiceps* cysticerci before the application of the carcinogen resulted in fewer tumors (*León-Cabrera et al., 2014*). Expression of both β-catenin, which plays a role in cell proliferation, and CXCR2, a neutrophil chemokine receptor (interleukin 8 receptor β), have been suppressed in CAC mice infected with *T. crassiceps*; elevated levels of these markers are associated with bowel cancer (*Ou et al., 2019*). An increase in the macrophage population and locally high levels of IL-4 in the intestinal tissue have also been observed during *T. crassiceps* infection (*Ledesma-Soto et al., 2015*). Excretory-secretory products of *T. crassiceps* larvae display a similar anti-carcinogenic effect when administered post-CAC induction (*Callejas et al., 2019*). Since the development of CAC is associated with an inflammatory environment, the tumor-suppressing effect of *T. crassiceps* could be linked with its ability to shift the immune response towards the anti-inflammatory Th2 (*Terrazas et al., 1998*).

GK-1, a peptide isolated from *T. crassiceps* larvae, could contribute to the anti-tumor effect. The synthetically generated GK-1 administered to breast cancer-bearing mice induced increased necrosis of tumors and reduction of lung metastases (*Torres-García et al., 2017*); the same effects were observed in mice with subcutaneous B16F10 melanomas (*Pérez-Torres et al., 2013*).

GK-1 also increases the efficiency of the dendritic cell mouse vaccination system to activate specific anti-tumor T cells. The most substantial effect on melanoma tumor reduction and mouse survival was observed when dendritic cells were pre-stimulated with GK-1 (*Piñón-Zárate et al., 2014*). The anti-tumor effect was strengthened by combining GK-1 with an anti-PDL1 ("programmed cell death ligand 1") antibody. PDL1 can be expressed on the surface of tumor cells (*Keir et al., 2008*), thereby reducing the cytotoxic effect of T cells on tumor cells by binding their receptor and inducing T cell apoptosis (*Iwai et al., 2002*). Mice treated with GK-1 and anti-PDL1 antibodies developed smaller B16F10 subcutaneous tumors and had their survival improved (*Vera-Aguilera et al., 2017*). GK-1 alone applied to mice with established B16F10 melanomas suppressed tumor growth. There was also an increase in the population of tumor-infiltrating CD8+ T cells, with decreased PD1 (receptor for PDL1) expression in these CD8+ T cells. Reducing PD1 expression in activated lymphocytes could increase their cytotoxic activity and lead to the observed suppression of tumor growth (*Rodríguez-Rodríguez et al., 2020*).

### Other tapeworms

*Hymenolepis nana* is another tapeworm that has been observed to inhibit tumor growth. Although this is the most prevalent tapeworm in the human intestine (adult forms), it is scarce as an extraintestinal larval infection in humans (*Olson et al., 2003*). In a carcinogen-induced skin tumor model, mice pre-infected with *H. nana* developed fewer tumors compared to controls. Furthermore, infected mice showed an increased number of eosinophils and neutrophils; the increased number of eosinophils could reduce the number of tumors (*Ramos-Martínez et al., 2019*). On the other hand, the related species, *Hymenolepis diminuta*, did not affect the C3(1)-TAg mouse model of breast cancer (*Sauer et al., 2021*).

**Table 1 Summary of tapeworm products suppressing cancer development.**

| Tapeworm | Developmental stage/Antigen | Assay | Effect | Cancer type (Abbreviations of cancer cell lines) | References |
|---|---|---|---|---|---|
| *E. granulosus* | Living larvae | *In vitro* | Cell lysis, proliferation inhibition | WEHI-164 fibrosarcoma, BHK fibroblasts | *Darani et al. (2012)* |
| | Living larvae | Rats | Reduced tumor growth | DMBA-induced breast cancer | *Altun et al. (2015)* |
| | Cystic fluid | Mice | Reduced tumor growth, NK cell activation | B16F10 melanoma, CT27 colon carcinoma, LL/2 lung cancer | *Darani et al. (2016)*, *Berriel et al. (2021)* |
| | 78 kDa fraction of cystic fluid | *In vitro* | Increased apoptosis | 4T1 breast cancer | *Daneshpour et al. (2019)* |
| | 78 kDa fraction of cystic fluid | Mice | Reduced tumor growth | B16F10 melanoma | *Rad et al. (2018)* |
| | miRNA-365 cystic fluid component | *In vitro* | Increased apoptosis | A375 melanoma | *Mohammadi et al. (2021)* |
| | EgKI-1 | Mice | Inhibited neutrophil chemotaxis | MDA-MB-231 breast cancer | *Ranasinghe et al. (2018)* |
| | EgKI-1 | *In vitro* | Inhibited proliferation and migration | MDA-MB-231, HeLa cell line | *Ranasinghe et al. (2018)* |
| | EgKI-4 (Kunitz4) | *In vitro* | Increased apoptosis, inhibited proliferation | HT29 colorectal adenocarcinoma, HepG2 liver cancer | *Rashno et al. (2023)* |
| | Egmuc | Mice | Increased cytotoxic effect of splenocytes | Panc02 pancreatic cancer | *Noya et al. (2013)* |
| *T. crassiceps* | Living larvae/ESP | Mice | Reduced tumor growth, Th2 polarization, reduction of b-catenin and CXCR2 expression | Colitis-associated carcinoma | *León-Cabrera et al. (2014)*, *Callejas et al. (2019)* |
| | GK-1 | Mice | Reduced tumor growth and metastases, increased numbers of cytotoxic CD8+ T lymphocytes | 4T1 breast cancer, B16F10 melanoma | *Torres-García et al. (2017)*, *Rodríguez-Rodríguez et al. (2020)* |
| *H. nana* | Living mature tapeworms | Mice | Reduced tumor growth, increased numbers of eosinophils and neutrophils | 7,12 dimethylbenz-anthracene-induced skin cancer | *Ramos-Martínez et al. (2019)* |
| *M. corti* | Living larvae | Mice/ *in vitro* | Activated eosinophil-induced apoptosis | A20 lymphoma | *Costain et al. (2001)* |
| *T. solium* | Calreticulin | *In vitro* | Reduced ability to form colonies, reduced viability | MCF7, SKOV3 adenocarcinoma | *Schcolnik-Cabrera et al. (2020)* |

**Note:**
ESP, excretory-secretory products; GK-1, peptide isolated from *T. crassiceps* larvae; EgKI-1, Kunitz-type protease inhibitor produced by *E. granulosus*; EgKI-4 (Kunitz4), Kunitz-type protease inhibitor produced by *E. granulosus*; Egmuc, mucin-like peptide isolated from *E. granulosus* larvae.

Although the effect of *Mesocestoides corti* on cancer was never directly studied, in one case, *M. corti* infection was used in mice to induce eosinophilia. Such activated eosinophils were isolated and stimulated apoptosis in a lymphoma cell line *in vitro* (*Costain et al., 2001*). Furthermore, *M. corti*, similar to *E. granulosus*, expresses the Tn antigen found on some cancer cells (*Ubillos et al., 2007*; *Medeiros et al., 2008*). To support the data mentioned above, our experimental results with *M. corti* and *T. crassiceps* show that

tapeworm larvae inhibit melanoma development in experimental mice (*Schreiber et al., 2024*).

Infection with *Taenia solium* is more often associated with cancer promotion (*Del Brutto et al., 1997*; *Herrera et al., 2000*); however, a recombinant form of calreticulin isolated from *T. solium* larvae has been shown to reduce the viability of the MCF-7 breast cancer cell line and the ability to form colonies of the SKOV-3 ovarian cancer cell line *in vitro* (*Schcolnik-Cabrera et al., 2020*).

## CONCLUSIONS

The interaction between tapeworms and cancers has yet to be fully understood. Based on the summary above, the tapeworms (like all other parasites/pathogens) might exhibit a "dual" role concerning particular cancer type development and parasite-cancer interaction. The limitations are based on a few tapeworm models used to test their effects on cancers and the unbalanced amount of data available (often, only *E. granulosus* has been studied in this regard, and any experimental data with humans are missing; only epidemiological surveys could indicate the interaction). Moreover, the observed effects are frequently related to a specific tapeworm-cancer combination. In some cases, particular authors have obtained contradictory effects (*e.g.*, inhibition *vs.* stimulation of cancer growth by *E. granulosus* (*Wang et al., 2009*; *Noya et al., 2013*)).

Our review shows that mainly tapeworm larvae invading tissues were studied in the past. The adult worms in the intestine were usually ignored (although they could influence the gut microbiota composition and general immune status (*Walusimbi et al., 2023*)). If some tapeworm larvae/larval products lead to the reduction or elimination of primary cancers and metastases, the molecular mechanisms behind the process should be studied and tested for their potential medical application (as a preventive or curative tool). Such activities are currently linked to the basic research stage (*in vivo* experiments with animals or *in vitro* tests with cancer cells); we did not record any clinical trial using tapeworms as an effector tool to suppress cancer.

Tapeworm larvae are generally pathogenic, administering a cocktail of bioactive tapeworm molecules (see Table 1 for example) seems like a more relevant approach. At least three groups of such molecules can be considered: (a) In some combinations, tapeworm larvae and cancers share antigenic epitopes (frequently glycans) that can cross-react with specific antibodies. This way, timely immunization with tapeworm antigens (which seem more immunogenic than the shared cancer antigens (*Berriel et al., 2021*; *Yousefi et al., 2023*) could protect against developing cancers in specific combinations. (b) Some tapeworm products could boost various (innate or adaptive) components of the immune system, which could then eliminate the cancer cells. The most probable mechanism is the activation of tumor-infiltrating NK cells or specific anti-tumor CD8+ T cells (*Rodríguez-Rodríguez et al., 2020*; *Berriel et al., 2021*). (c) It seems that some parasite-derived molecules directly affect cancer cells (their activities, including replication, migration, invasiveness, apoptosis, *etc.*), and they are effective in the *in vivo* system (*Ranasinghe et al., 2015*, *2018*). In all these possible effects of tapeworm products, the mode of administration into the host body will play a crucial role and must be tested.

To conclude, the prospect of bioactive tapeworm molecules helping reduce the growth and metastasis of cancer is within the realm of future possibility, although extensive research is yet required due to certain concerns, which we already mentioned above. Furthermore, these tapeworms may cause life-threatening tissue helminthoses, besides being the source of substances with potential anti-cancer properties. In order to avoid infection with living parasites by utilizing said molecules directly, the effectors' production, their formulation, and administration needs to be managed first, in order to test the desired effects, while the precise mode of action of these compounds must also be characterized.

### Funding
This research is currently supported by the Czech Science Foundation (21-28946S), the European Regional Development Fund and Ministry of Education, Youth and Sports of the Czech Republic (CZ.02.1.01/0.0/0.0/16_019/0000759), the Charles University Grant Agency (B-BIO 283823), and the Charles University institutional support (Cooperatio Biology, UNCE/SCI/012-204072/2018, UNCE24/SCI/011, and SVV 260678/2023).
The funders had no role in study design, data collection and analysis, decision to publish, or preparation of the manuscript.

### Grant Disclosures
The following grant information was disclosed by the authors:
Czech Science Foundation: 21-28946S.
European Regional Development Fund and Ministry of Education, Youth and Sports of the Czech Republic: CZ.02.1.01/0.0/0.0/16_019/0000759.
Charles University Grant Agency: B-BIO 283823.
Charles University institutional support: Cooperatio Biology, UNCE/SCI/012-204072/2018, UNCE24/SCI/011, and SVV 260678/2023.

### Competing Interests
The authors declare that they have no competing interests.

### Author Contributions
- Manfred Schreiber conceived and designed the experiments, performed the experiments, analyzed the data, prepared figures and/or tables, and approved the final draft.
- Vojtěch Vajs performed the experiments, prepared figures and/or tables, and approved the final draft.
- Petr Horák conceived and designed the experiments, authored or reviewed drafts of the article, and approved the final draft.

### Data Availability
   This is a literature review.

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
