# Peer review of "How tapeworms interact with cancers: a mini-review"

_PeerJ, doi:10.7717/peerj.17196_

## Round 0.1 · original submission · Major Revisions

A review of the potential tapeworm influence on host response to cancer is timely and of broad relevance. As far as I can tell from a literature search and supported by the assessment of the reviewers, this subject was neglected in previous reviews. However, there are some crucial aspects which need to be addressed before publication. The main points to be addressed are:

Please state and explain how systematic your review is: I agree with reviewers 1 and 2 (even in case of a more narrative review) that you need to specify more clearly how the studies you included were searched/found/selected, what criteria were used to include or exclude individual studies and how the quality of individual studies were assessed. In case of a systematic review (compare reviewer 1), even more stringent criteria need to be applied (e.g., PRISMA checklist and chart, etc.). Although in your purpose you list epidemiological, experimental and molecular evidence in additional to cellular data, relevant details and studies are missing (compare reviewer 1). A flow chart would be beneficial to clarify your approach to the reader (compare reviewer 2). Please state why infection with E. multilocularis seem to have been excluded from your review (compare reviewer 2). Please state more explicitly why you focused on studies between 1975-2023 (compare reviewer 2)?

Justify more clearly why immune responses triggered by helminth infections were excluded: stating them to be non-relevant is not sufficient as immune response modulated by helminth infections have crucial roles in response to malignancies as highlighted by reviewer 1

Add relevant missing studies: as pointed out by reviewer 1 several key works seem to be missing related to your listed aim(s). It is crucial that the literature is reviewed comprehensively and systematically and the reasons for inclusion/exclusion to be reproducible, homogenous and explicitly stated.

Please standardize the use of abbreviations: I agree with reviewer 2 that abbreviations should be written out and/or explained, the first time they are used
Please identify recommendations for future directions: I agree with reviewer 1 that it would be crucial to add recommendations for future research and the knowledge gaps as well as the limitations of research on this topic (compare also the guidelines for reviews which state that unresolved questions / gaps / future directions should be identified in the conclusions).
Referencing and formatting of table and figures: please make sure the tables and figures are properly referenced in text (compare reviewer 2). Please check formatting of the table 1 and make sure the abbreviations in the heading of table 1 are written out in full (compare reviewer 2).

Please address these points and all other points raised in the reviews.

I look forward to receiving the revised manuscript.

·

Basic reporting

How tapeworms interact with cancers: a minireview

The manuscript presents a brief account on different aspects of the interactions between the tapeworms and cancer. The authors have tried to review tapeworms' influence on the host response to malignancies. The topic is very interesting and important, however the various aspects of cestode-cancer interactions are not covered in the review.
According to the authors the purpose of the review is to summarize the epidemiological, cellular, molecular and immunological data on the "cancer-helminth co-occurrence" and to review the in vitro and in vivo data of the effects of cestode products and molecules on cancer cells, however the manuscript is too short to give an overall big picture of the subject to the readers. Details have not been given on the epidemiological, experimental and molecular evidences and the knowledge gaps on this issue. The following issues should be addressed in the manuscript:

- First of all it is not clear if the review is a systematic or a narrative review. If the manuscript presents a systematic review then many essential elements of the systematic review are absent in the study, e.g. PRISMA checklist and chart, detailed inclusion/exclusion criteria, quality assessment, etc.

- line 95-96. The authors exclude the articles covering immune responses triggered by helminth infections because they are "non-relevant". As we know (and the authors also pointed out in the MS) immune responses modulated by helminth infections have crucial roles in the response to malignancies.

- Several key works on this topic have been missed in the review. for example a cardinal work published in the New England Journal of Medicine on the correlation of Hymenolepis nana and malignancy (Muehlenbachs A et al, N Engl J Med, 2015, 373;19. PMID: 26535513) is not discussed in the review. Plus other works: Paul Brindley: PMID: 26618199 Why Does Infection With Some Helminths Cause Cancer? Trends Cancer. 2015;1(3):174-182. AND Rashno et al. Design of ion channel blocking, toxin-like Kunitz inhibitor peptides from the tapeworm, Echinococcus granulosus, with potential anti-cancer activity." Scientific Reports 13.1 (2023): 11465. and a couple of works on Sparganosis, Spirometra spp. etc.

- Please provide recommendations for future research and the knowledge gaps as well as the limitations of research on this topic.

Thank you.

Experimental design

Please see above.

Validity of the findings

Please see above.

Additional comments

Please see above.

Cite this review as

·

Basic reporting

The review on "How tapeworms interact with cancer" is well written, based on an extensive literature review and discusses the pros and cons of the possible involvement of parasites in the development or suppression of cancer.

The review is worth to be published after a small revision.

Minor comments:

Title: How does the "mini Review" come about, with 13 pages of text and 75 literature citations?

Abstract:
a. Please write more precisely where 19.3 million new cases of cancer occur each year (worldwide? I found it in the WHO report) or better what the incidence per 100,000 inhabitants per year is worldwide. State this in the same way for infection-associated oncological diseases.
b. Write a "Conclusio" in the last paragraph, this is missing in the abstract.

Main text:
Abbreviations in the text are partly written out and or explained the first time, do not do e.g. WHO, TH2, Treg Tag.....please standardize both

Table 1 is a compact and comprehensible summary:
a. Table 1 is only referred to once in the text, with Table 1 summarizing all experiments.
b. In the first row of the table, the 2nd column should be labeled after "Tapeworm", e.g. developmental stage/antigen.
c. In the heading of Table 1, the abbreviations in the table should be written out in full.

Figure 1 is understandable and clear:
a. The figure is not referred to in the text (?)
b. In the caption of Figure 1, the abbreviations in the figure should be written out in full.

Experimental design

The literature survey for the review includes articles published in the period 1975-2023 with the terms "Tapeworm", "Cestod", "Cancer", "Taenia", "Echinococcus", "Mesocestoides" and "Hymenolepis".

a. Why this time limit? Were there no articles about this topics before?
b. Were articles with obsolete or alternative terms such as "echinococcosis", "hydatidosis" or "cystic hydatid disease" also considered?
c. Why was only the infection with E. granulosus and not also E. multilocularis considered for echinococcosis?
d. I would recommend a flowchart of how the literature was chosen and selected. The inclusion and exclusion criteria should be listed in detail. How many in-vitro, in-vivo (animal studies and human studies e.g. epidemiologic studies and observational studies)
e. What type of studies, and how the quality of a study was assessed? 44 studies were included, but how many studies were excluded and why?

Validity of the findings

The report deals with current and important data and the topic is still neglected.

Additional comments

no comment

---

## Round 0.2 · Minor Revisions

The changes to the revised manuscript make it easier to follow, reproduce and of broader relevance. I would like to see it published as it will be of broad relevance to the community interested in cancer-tapeworm interactions, but some minor but crucial points related to formatting need to be addressed before publication:

1) Title: Given it focus on particular kind of interactions as you clearly and comprehensively describe in the manuscript, i would like to see the term mini-review in the title as it was during the initial submission.

2) Use of developing countries: the use of the term is a bit out-dated and it has been recommended to phase out its use by most world organizations (e.g., World Bank). It would therefore be better to refer to the lower to middle-income countries as the Global South or at least exchange developed by "high income" (see suggestions in annotated pdf)

3) Formatting: please do repeat the authors within brackets with year (only the year is necessary). It concerns Line 122: "Guan et al. (Guan et al. 2019)" as well as Line 142: "Gao et al. (Gao et al. 2018)"

4) Reference to unpublished observations: please refer to unpublished observations as follows - list all author names, followed by ", personal observations, unpublished). If it concerns an accepted manuscript or published pre-print, please cite the full details.

5) Table: I agree with reviewer 1 that Kunitz4 (EgKI-4) should be added to the table. Only Kunitz1 is so far added here.

Please addressed all these points - also those listed in the annotated pdf.

I look forward to receiving the revised manuscript.

·

Basic reporting

Most of the comments are considered in the revised version of the manuscript. In Table 1, please add Kunitz4 (EgKI-4) to the Table. Thanks.

Experimental design

no comment.

Validity of the findings

no comment.

Additional comments

Most of the comments are considered in the revised version of the manuscript. In Table 1, please add Kunitz4 (EgKI-4) to the Table. Thanks.

Cite this review as

·

Basic reporting

no comment

Experimental design

no comment

Validity of the findings

no comment

Additional comments

The authors have made all necessary changes and answered open questions. The work can therefore be accepted

---

## Round 0.3 · accepted · Accept

Thank you for implementing these additional suggestions to the text and table. I feel the paper is now ready be published. Please consider using "high-income global north countries with high hygiene standards" or "high-income countries with high hygiene standards in the global north" for the sake of completeness and not implying that this only relates to income and hygiene as climate/latitude/geographic position may also play a role. I look forward to seeing this mini-review published.